# Superacid-resistant macrocyclic BODIPYs

Keita Watanabe[1,5], Gentaro Honda[1,5], Yuki Terauchi[1], Shunsuke Mamiya[1], Yuya Inaba[1], Tasuku Nakajima[2,3], Jian Ping Gong[2,3], Yusaku Yamaguchi[1], Yuichi Kitagawa[1,2], Yasuchika Hasegawa[1,2], Yuki Ide[1,2], Min Gao[2], Tomoki Yoneda[1,4] ✉ & Yasuhide Inokuma[1,2] ✉

Boron-dipyrromethenes (BODIPYs) are versatile fluorophores with intense fluorescence and broad applications in bioimaging and sensing. However, they undergo deboronation under acidic conditions, which causes fluorescence degradation. Herein, we designed exceptionally acid-stable BODIPYs by harnessing the synergistic boron-chelation effect of calix[3]pyrrole-like macrocycles. We show that their characteristic optical properties are retained in strongly acidic media, including superacids, without undergoing deboronation. Macrocyclic BODIPYs exhibit sharp absorption and protonation-induced fluorescence switching, with quantum yields of up to 0.90 and narrow Stokes shifts. Notably, no deboronation was observed even in non-diluted fluorosulfuric acid, and visible fluorescence was sustained for over a day. Beyond their unusual acid resistance, the macrocyclic BODIPYs had higher thermal- and photostability compared with conventional BODIPYs. Peripheral substitution allowed the modulation of absorption and emission wavelengths, and fluorous-tagging through axial ligand exchange enabled fluorescence switching in response to perfluorooctanoic acid in a fluorous solvent. We used superacid-resistant BODIPYs as acid indicators for the fluorescence staining of Nafion beads and sulfonylated gels, which are too acidic to sustain the fluorescence emission of conventional BODIPYs. Our findings expand the scope of BODIPYs into strongly acidic, non-aqueous environments, opening opportunities for fluorescence imaging and sensing in materials and biological systems.

Boron-dipyrromethene (BODIPY) dyes[1] have firmly established themselves as biological markers[2-4], chemosensors[5,6], and photosensitizers[7,8] owing to their distinctive optical properties, i.e., narrow Stokes shifts and high-emission quantum yields. Since their absorption/emission wavelengths can be tuned by peripheral substitutions[9], various BODIPY analogs have been used to stain cell proteins, lipids, and metabolites[10,11], and to detect molecules and ions[12,13]. However, a common and critical drawback of BODIPYs is that they readily release the boron atom in the presence of Brønsted acids,

resulting in fluorescence degradation (Fig. 1)[14-16]. Consequently, their usefulness in acidic media is limited. Moreover, although several acid-stable BODIPY analogs have been developed for use in aqueous solutions down to pH ~ 1[17], their deployment in strongly acidic and non-aqueous environments remains particularly challenging owing to rapid acid-induced deboronation.

Among pyrrole-based boron complexes, boron(III)-subphthalocyanines (SubPcs) and -subporphyrins (SubPors)[18-21] are remarkably resistant to acid-mediated deboronation. Under acidic

[1]Division of Applied Chemistry, Faculty of Engineering, Hokkaido University, Sapporo, Japan. [2]Institute for Chemical Reaction Design and Discovery (WPI-ICReDD), Hokkaido University, Sapporo, Japan. [3]Faculty of Advanced Life Science, Hokkaido University, Sapporo, Japan. [4]Department of Pharmaceutical Sciences at Narita, International University of Health and Welfare, Narita, Japan. [5]These authors contributed equally: Keita Watanabe, Gentaro Honda. ✉e-mail: t-yoneda@ihwg.jp; inokuma@eng.hokudai.ac.jp

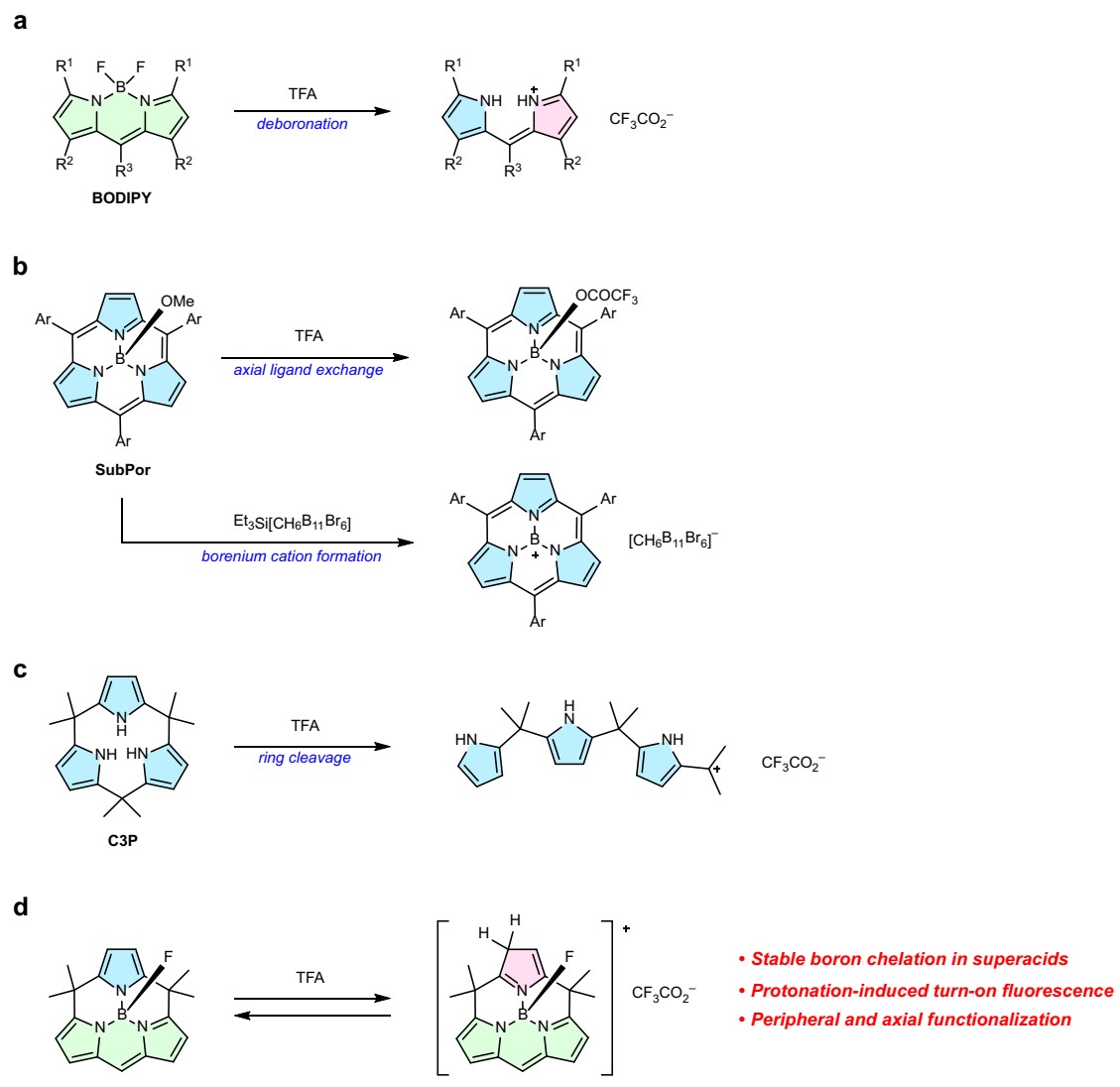

**Fig. 1 | Acid responses of pyrrole-based compounds. a** Typical deboronation reactions of conventional BODIPYs in the presence of Brønsted acids, such as trifluoroacetic acid (TFA). **b** Acid-stable chelation of the boron atom in boron(III)−SubPor. **c** Brønsted acid-induced ring cleavage reaction of C3P. **d** The tripyrrolic macrocycle developed in this study retains boron chelation even after protonation and exhibits characteristic BODIPY fluorescence.

conditions, protonation or Lewis acid coordination at the *meso*-azomethine units is commonly observed for subphthalocyanines[22], whereas acids often promote boron-axial ligand exchange at the boron center of hydroxo- or alkoxo-substituted subporphyrins[23]. Nonetheless, deboronation yielding free-base SubPcs or SubPors has not been observed to date. Even upon treatment with a strong Lewis acids, such as a silylium reagent, boron chelation is maintained as the coordinatively unsaturated borenium cation forms[24,25]. While SubPcs and SubPors use a 14π-aromatic macrocycle for boron chelation, a similar chelation effect has recently been observed in a boron(III) complex of calix[3]pyrrole (C3P), which has a globally non-aromatic macrocycle[26,27]. C3P is a porphyrinogen-like tripyrrolic macrocycle that undergoes acid-mediated macrocyclic ring cleavage in its boron-free form[28,29]. In contrast, the boron(III)-C3P (**1**) resists both macrocyclic ring cleavage and deboronation in the presence of trifluoroacetic acid, whereas most BODIPYs release the boron atom under similar conditions (Fig. 1). Given the synergistic relationship between the boron(III) atom and C3P, we envisioned that an in-depth understanding of the crucial effect of tripyrrolic macrocycle **1** on boron(III)-chelation would provide a design standard for acid-tolerant BODIPY dyes. Herein, we report the amphoteric nature of boron(III)-

C3P **1**, which maintains stable chelation of the boron atom under strongly acidic conditions in exchange for protonation-induced dearomatization of the pyrrole units. Notably, this acid tolerance is associated with the formation of a tetracoordinated boronium cation species, in contrast to the borenium cation-based stabilization observed in SubPcs and SubPors. We exploited this phenomenon to develop exceptionally acid-stable macrocyclic BODIPYs that retain the boron atom even in concentrated superacids, such as trifluoromethanesulfonic acid or fluorosulfonic acid, and exhibit fluorescence switching upon protonation. Boron axial ligand exchange and peripheral substitution reactions can change their solubility and absorption/emission wavelength of resulting BODIPYs. The superacid-stable BODIPY analogs complement the poor acid stability of conventional BODIPYs and enable fluorescence staining of various acidic media, including Nafion, sulfonated organogels, and cation exchange resins, as acid indicators.

## Results

### Boron(III)−tripyrrolic macrocycle synergy for acid stability

To explore the origin of the robust boron chelation observed in boron(III)-C3P **1** under acidic conditions, we examined its solution

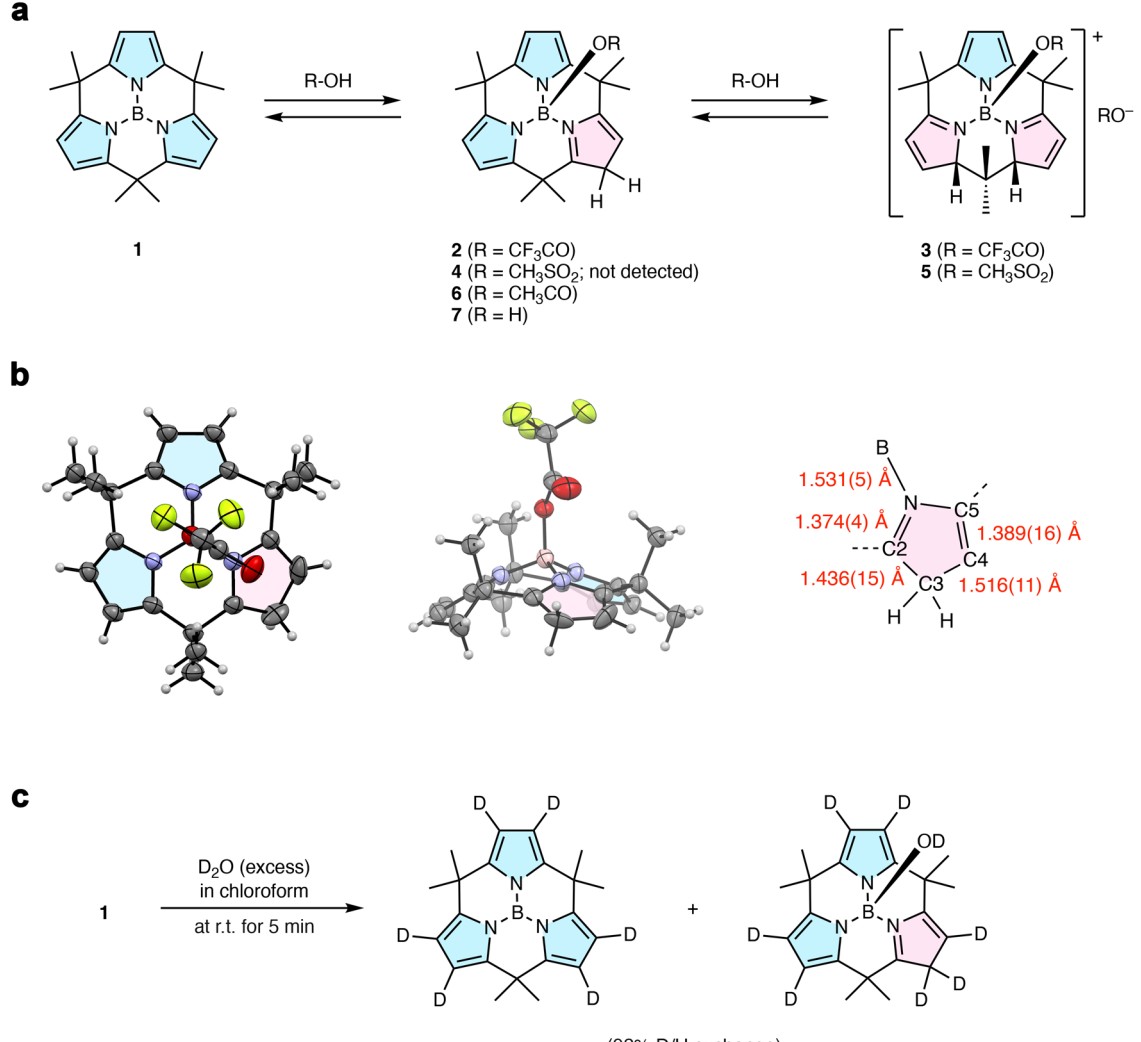

**Fig. 2 | Protonation behavior of boron(III)−C3P complex. a** Protonation equilibrium of compound **1** in solution as observed by NMR analyses. **b** Single crystal X-ray structure (left: top view; middle: side view) of TFA adduct **2**, and bond lengths around the 3*H*-pyrrole ring (right). Thermal ellipsoids are set at the 50% probability. **c** Proton–deuterium exchange reaction of **1** under neutral conditions.

behavior in the presence of trifluoroacetic acid (TFA). The proton nuclear magnetic resonance (¹H NMR) spectrum of **1** recorded in dehydrated CDCl₃ (at 6.0 mM) revealed only two singlet signals at 5.95 and 1.69 parts per million (ppm), which were assignable to pyrrole *β*-CH and methyl protons, respectively. The ¹¹B NMR signal of **1** at 22.81 ppm indicated a trigonal-planar boron(III) center, as confirmed by its single crystal X-ray structure. When 3 equivalents of TFA were added to a solution of **1** at 25 °C, a new set of proton signals attributable to the TFA-adduct **2** appeared in a molar ratio of **1:2** = 1/2. Adduct **2** was formed through an equilibrium process; whereas the signal ratio of **2** decreased with increasing temperature (51% and 40% at 40 and 50 °C, respectively), **2** was predominant (>90%) at 0 °C (Supplementary Fig. 2). Adduct **2** showed geminally coupled methylene proton signals at 3.94 and 3.68 ppm along with six different methyl protons and five pyrrole *β*-CH signals, reflecting its $C_1$ molecular symmetry. The ¹¹B NMR signal of **2** was observed at −2.18 ppm, indicating a tetrahedrally coordinated boron center.

Slow evaporation of the solvent from a CH₂Cl₂/TFA solution furnished diffraction-grade single crystals of **2**. As indicated by NMR spectroscopy, the crystal structure of **2** revealed an axially trifluoroacetoxy-bound boron center in a tetrahedral coordination geometry (Fig. 2). One of the three pyrrole units had a significantly

longer N−B bond length (1.531(5) Å) than the others (1.490(4)–1.493(4) Å). The N−C2, C2−C3, C3−C4, and C4−C5 distances of this pyrrole ring were 1.374(4), 1.436(15), 1.516(11), and 1.389(16) Å, respectively, clearly indicating a 3*H*-pyrrole-type bond length alternation. These structural analyses showed the amphoteric nature of **1** during TFA addition; namely, the pyrrole unit behaves as a base to be protonated, and the three-coordinated boron center acts as a Lewis acid.

In the presence of 25 equivalents of TFA, **2** underwent further protonation to yield boronium cation **3** exclusively at room temperature. The ¹H NMR spectrum of **3** recorded in CDCl₃ featured a $C_s$-symmetric signal pattern with three pyrrolic *β*-proton signals at 8.14, 7.16, and 6.08 ppm as a couple of doublets (*J* = 5.5 Hz) and a singlet, respectively. In addition, a broad signal attributable to the pyrrole *α*-CH was observed at 5.90 ppm. The ¹¹B NMR signal at −2.81 ppm suggested the presence of a tetracoordinated boron center. Further heteronuclear multiple bond coherence measurements revealed the structure of **3** (Supplementary Fig. 7). While **2** contained a 3*H*-pyrrole moiety, the doubly protonated species **3** exhibited two 2*H*-pyrrole rings resulting from *α*-protonation. Density functional theory (DFT) calculations also supported that structure **3** was more stable than the other structural isomers, namely, the *β*,*β*- and *α*,*β*-protonated forms (Supplementary Table. 1).

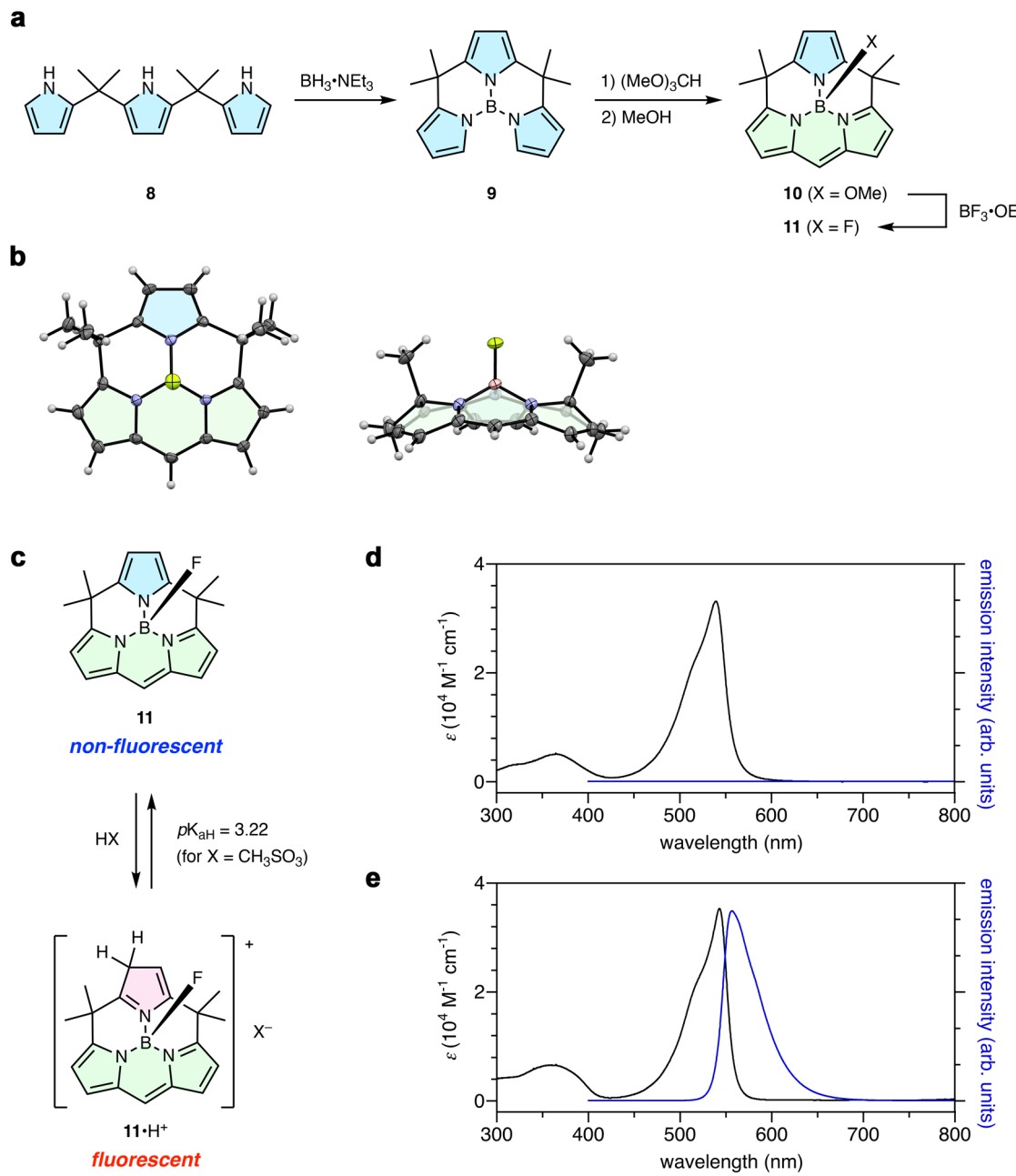

**Fig. 3 | Synthesis and acid-responsive optical properties of macrocyclic BOD-IPY 11. a** Synthetic route to compound **11**. **b** Top and side views of the crystal structure of **11**. **c** Protonation equilibrium between non-fluorescent **11** and emissive **11·H⁺**. **d** UV–vis absorption (black) and fluorescence emission (blue) spectra of **11** in dichloromethane. **e** UV–vis absorption (black) and fluorescence emission (blue) spectra of **11·H⁺** in dichloromethane containing 5 vol% TFA.

When 1 equivalent of methanesulfonic acid (MSA) was added to **1** in CDCl₃, the expected monoadduct **4** was not detected by ¹H NMR spectroscopy even at −60 °C. Instead, the doubly protonated species **5** was observed in the equilibrated mixture. A characteristic proton signal at 6.14 ppm indicated the presence of 2H-pyrrole rings in **5**. Adduct **5** was sufficiently stable to retain both the boron atom and the macrocyclic framework even in neat MSA. When acetic acid (AcOH) was used to protonate **1**, the monoprotonated form **6** was observed below 0 °C, but boronium cation species such as **3** and **5** were not detected, even in the presence of a large excess (>100 equivalent) of AcOH. Similar addition reactions with less acidic water and methanol were also confirmed by NMR spectroscopy.

In a D₂O-saturated chloroform solution, **1** exhibited H/D exchange of the pyrrole β-protons through formation of hydrated form **7**. When excess D₂O was added to a CDCl₃ solution of **1**, the intensity of the pyrrole β-proton signal at 5.95 ppm was attenuated by 92% after 5 min at 298 K. In contrast, boron-free C3P exhibited H/D exchange only at the NH proton under the same conditions. The van't Hoff plot analysis of the equilibrium constants between **1** and hydrated **7** at various temperatures gave physical parameters of $\Delta H = -56.5\,\text{kJ mol}^{-1}$, $\Delta S = -151\,\text{J mol}^{-1}\,\text{K}^{-1}$, and $\Delta G = -11.46\,\text{kJ mol}^{-1}$ at 298 K. Although one of the pyrrole moieties is to be dearomatized, the favorable formation of **7** at room temperature is attributable to the enhanced basicity of the pyrrole unit upon coordination of the fourth ligand to the boron atom.

## Synthesis and properties of superacid-resistant macrocyclic BODIPYs

Inspired by the remarkable boron-chelating behavior of **1**, we designed an acid-resistant macrocyclic BODIPY framework, designated as boron(III)-calix[1]dipyrrin[1]pyrrole (compound **11**, Fig. 3). This tripyrrolic

macrocycle features a ring size and boron-chelation environment similar to those of C3P and SubPor, and incorporates dipyrromethene-type π-conjugation along with an isolated pyrrole unit. To construct macrocycle **11**, tripyrrane **8** was reacted with borane-triethylamine to give boron(III)-tripyrrane **9** in 98% yield[30,31]. Macrocyclization in excess trimethyl orthoformate afforded stable boron(III) complex **10** in 46% yield after treatment with methanol. Although boron complex **9** readily released the boron atom to give starting material **8** in the presence of water in chloroform, no deboronation was observed for macrocycle **10** under either aqueous conditions or in the presence of a Lewis acid. Since the axial-alkoxo group on the boron was susceptible to axial ligand exchange[27,32,33], fluoride ligand was introduced to give **11** as a more stable analog. Substitution of the fluoride was confirmed by a doublet signal at −2.69 ppm ($J_{B-F}$ = 40.0 Hz) in the $^{11}$B NMR spectrum (Supplementary Fig. 63). Single crystal X-ray diffraction analysis revealed that the central boron atom of **11** adopts a tetrahedral coordination geometry with an axially coordinated fluoro ligand (Fig. 3b). The tripyrrolic macrocycle acts as a dianionic tridentate ligand in a manner similar to SubPor, while its π-conjugation is split into dipyrromethene and pyrrole fragments by two $sp^3$-hybridized carbon atoms at the *meso*-positions.

Macrocyclic BODIPY **11** exhibited protonation behavior analogous to that of **1**. The $^1$H NMR spectrum of **11** in CDCl$_3$ at 6.0 mM showed a couple of doublets at 6.94 and 6.26 ppm and a singlet at 5.88 ppm, attributable to *β*-protons of the dipyrromethene and isolated pyrrole moieties, respectively. Upon addition of 20 equivalents of TFA, the singlet signal broadened, while the time-averaged $C_s$-symmetric signal pattern remained intact. However, desymmetrization of the signal pattern was observed at −50 °C, with a pair of doublets appearing at 4.44 and 4.16 ppm, assignable to the methylene protons of the 3*H*-pyrrole moiety in **11•H$^+$** (Supplementary Fig. 21). Similar spectral changes were also observed upon addition of MSA and trifluoromethanesulfonic acid (TfOH), without any loss of the boron atom. $^1$H NMR titration of **11** with MSA determined the acid dissociation constant of the protonated form **11•H$^+$** as $pK_{aH}$ = 3.22 in acetonitrile-$d_3$ (Supplementary Fig. 22).

Compound **11** exhibited acid-responsive, BODIPY-like visible fluorescence with a small Stokes shift. In dichloromethane, **11** showed the lowest energy absorption band at 540 nm with an absorption coefficient of $\varepsilon$ = 3.3 × 10$^4$ M$^{-1}$ cm$^{-1}$ due to the π−π* transition of the boron(III)-dipyrromethene moiety (Fig. 3d). However, no fluorescence emission was observed upon excitation at the lowest energy absorption band under neutral conditions. This is attributable to intramolecular photo-induced electron transfer (PET) from the pyrrole fragment to the dipyrromethene unit, as similar phenomena have been reported for BODIPY analogs bearing electron-donating moieties[34,35]. Upon addition of TFA, a dichloromethane solution of **11** emitted yellow fluorescence at 556 nm (Stokes shift: 397 cm$^{-1}$), whereas the absorption spectrum remained virtually unchanged (Fig. 3e). These spectral changes indicated the contribution of the protonated form **11•H$^+$**, in which the electron transfer from the 3*H*-pyrrole moiety to the BODIPY core is unlikely to occur. In dichloromethane containing 5 vol% TFA, the fluorescence quantum yield and emission lifetime of **11** were 0.86 and 10.9 ns, respectively.

The time-dependent DFT calculations together with electrochemical analysis based on the Rehm−Weller approximation supported the fluorescence switching behavior of the macrocyclic BODIPY system. In the neutral state **11**, the lowest-energy absorption band appeared at 443.2 nm, with an oscillator strength (*f*) of 0.3876, corresponding to a π−π* transition localized within the BODIPY core. This transition involved excitation from HOMO−1 to the LUMO. However, the HOMO was primarily localized on the isolated pyrrole unit, which acted as an electron donor, resulting in fluorescence quenching due to efficient non-radiative decay pathways (Supplementary Fig. 17). The protonated form **11•H$^+$** exhibited a comparable absorption band at

449.3 nm (*f* = 0.3281), which also corresponded to a π−π* transition within the BODIPY core. The protonated state exhibited a distinctive lowering of the energy level of the pyrrole unit below that of the HOMO of the BODIPY moiety, thus suppressing the intramolecular electron transfer from the pyrrole unit. Electrochemical analysis further supported the occurrence of PET of **11** under neutral conditions. The first oxidation and reduction potentials of **11** were observed at 0.70 and −1.53 V (vs. ferrocene/ferrocenium ion couple), respectively, by cyclic voltammetry in CH$_2$Cl$_2$ (Supplementary Fig. 26). The free energy change for the electron transfer process, estimated using the Rehm−Weller equation[36], was −0.46 eV, indicating that the PET process is thermodynamically favorable. In contrast, the first oxidation peak at 0.70 V was significantly attenuated in the presence of 5 v/v% TFA, presumably due to protonation at the pyrrole moiety. This observation is consistent with the emergence of fluorescence emission under acidic conditions, where protonation resulted in the weakening of the electron-donating character of the pyrrole moiety. The calculation and electrochemical analyses indicated that **11** functions as a turn-on type fluorescent BODIPY analog under acidic conditions, in which the pyrrole fragment is protonated.

Stability tests with various acids confirmed the exceptional superacid tolerance of macrocyclic BODIPY **11** against acid-induced deboronation. We measured the absorption and fluorescence spectra of **11** in various media, including strong acids, such as MSA and fluorosulfonic acid (Fig. 4a). The absorption spectra of **11** remained essentially unchanged, regardless of the solvent polarity and acidity. However, fluorescence emission was not observed in neutral and weakly acidic solvents, such as tetrahydrofuran, acetonitrile, dimethyl sulfoxide, and acetic acid (Supplementary Fig 24). In contrast, when the solvent acidity increased to a Hammett acidity function ($H_0$)[37] of ≤−2.2, **11** exhibited pronounced fluorescence attributable to the protonated form **11•H$^+$**. The fluorescence quantum yields in formic acid ($H_0$ = −2.2), TFA ($H_0$ = −3.0), 42% aqueous tetrafluoroboric acid ($H_0$ = −4.2), 55% aqueous hexafluorophosphoric acid, and MSA ($H_0$ = −7.9) were determined to be 0.46, 0.81, 0.56, 0.56, and 0.75 with corresponding Stokes shifts of 372, 371, 342, 307, and 338 cm$^{-1}$, respectively. Remarkably, intense fluorescence was retained even in concentrated sulfuric acid ($H_0$ = −12.0), chlorosulfonic acid ($H_0$ = −13.8) and fluorosulfonic acid ($H_0$ = −15.1), with fluorescence quantum yields of 0.85, 0.68, and 0.90, respectively, indicating that the boron center remains intact in superacids (Fig. 4b and Table 1). Notably, a fluorescence quantum yield of 0.90 was observed in fluorosulfonic acid 12 h after dissolving **11**, and continuous monitoring revealed that the intense fluorescence was maintained for over a day. It is noteworthy that the absorption spectra of **11** in these acids exhibited a shoulder around 550 nm. Matrix-assisted laser desorption/ionization-time-of-flight mass spectrometric analysis of a solution of **11** in fluorosulfonic acid revealed partial fluorosulfonylation of the BODIPY core as a side reaction. These results suggest that direct sulfonylation of **11** under harsh conditions using concentrated sulfonic acids can proceed without deboronation, whereas conventional BODIPYs typically require mild conditions for sulfonylation[38]. In trifluoromethanesulfonic acid (TfOH), **11** underwent multiple protonation at both the BODIPY core and the mono-pyrrole fragment, leading to sequential changes in the absorption and fluorescence spectra depending on the TfOH concentration (Supplementary Fig. 25). Despite the harsh acidic conditions, boron complex **11** was recovered upon neutralization of the solution, demonstrating the exceptional superacid resistance of the BODIPY core to deboronation.

To demonstrate the enhanced acid stability of **11**, we performed comparative experiments with conventional BODIPY **12** and B−CN analog **13** (Fig. 4c, Supplementary Fig. 27), which has been reported to exhibit notable acid resistance[39–41]. While most reports on acid-tolerant BODIPYs have evaluated fluorescence emission in aqueous media with pH ≥ 0, macrocyclic BODIPY **11** remained fluorescent even at pH = −1.1

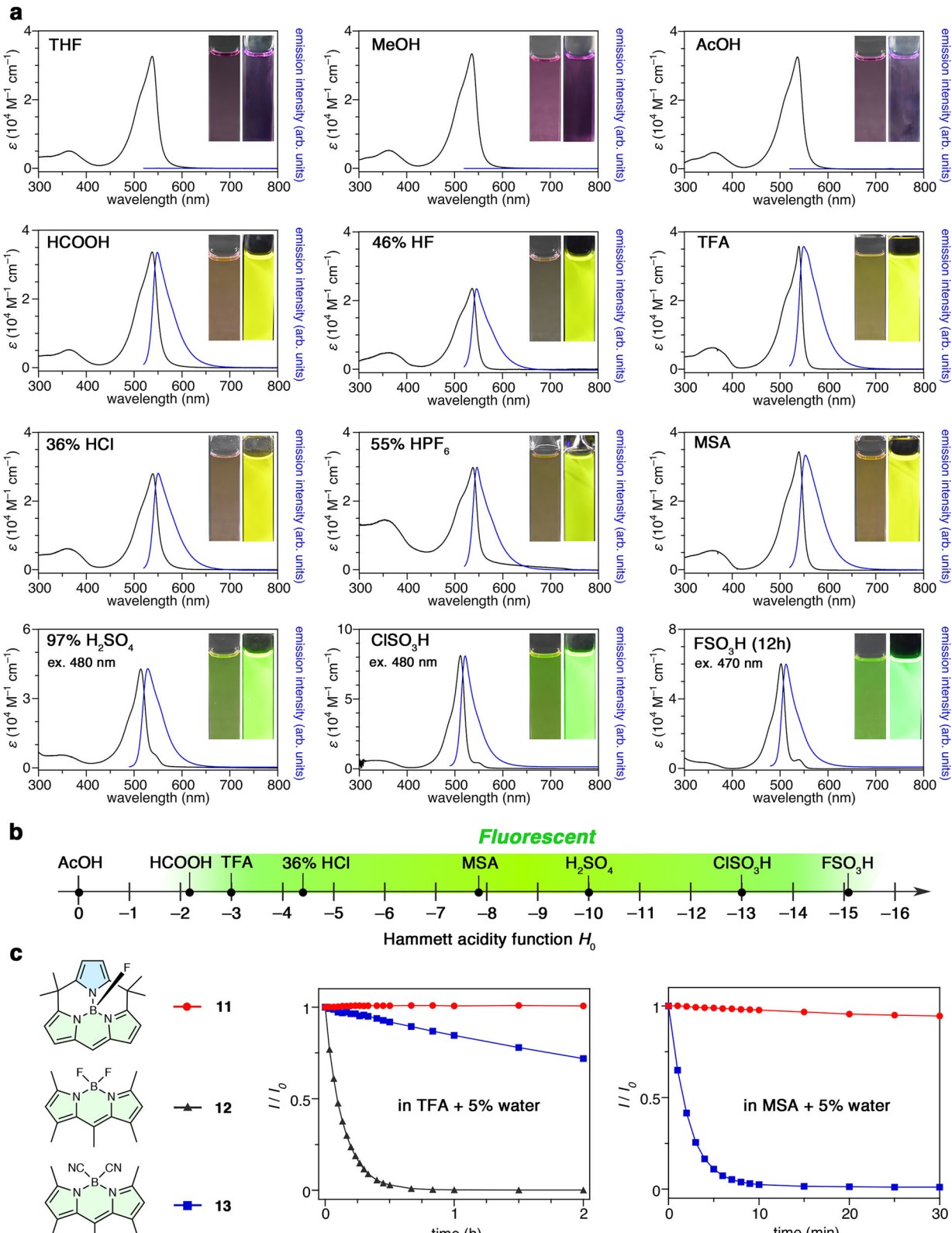

**Fig. 4 | Superacid resistance of macrocyclic BODIPY. a** UV–vis absorption (black) and fluorescence emission (blue) spectra of **11** in various acids. Unless otherwise noted, the fluorescence spectra were recorded with excitation at 510 nm. Insets show the solution colors under ambient light (left) and UV light at 352 nm (right). **b** Fluorescence emission window of **11** along the Hammett acidity function. **c** Time-dependent fluorescence intensity changes of BODIPY analogs **11, 12**, and **13** in (left) TFA with 5% water and (right) MSA with 5% water. *I/I_O* denotes the fluorescence intensity normalized to the initial value.

**Table 1 | Photophysical properties of 11 in various neutral solvents and acidic media**

| Solvent/acid | $\lambda_{abs,max}$ (nm) [$\varepsilon$ ($10^4$ M$^{-1}$cm$^{-1}$)] | $\lambda_{flu,max}$ (nm) [$\Phi_F$ (%)][a] | pKa (DMSO) | Hammett acidity function $H_0$ |
|---|---|---|---|---|
| Dichloromethane | 540 [3.33] | n.d. [<0.01] | – | – |
| Acetonitrile | 535 [3.23] | n.d. [<0.01] | 31.3 | – |
| Tetrahydrofuran | 538 [3.26] | n.d. [<0.01] | – | – |
| Dimethyl sulfoxide | 537 [2.99] | n.d. [<0.01] | 35.1 | – |
| 2-Propanol | 537 [3.14] | n.d. [<0.01] | 30.3 | – |
| Ethanol | 536 [3.17] | n.d. [<0.01] | 29.8 | – |
| Methanol | 536 [3.33] | n.d. [<0.01] | 29.0 | – |
| Trifluoroethanol | 536 [2.93] | n.d. [<0.01] | 23.5 | – |
| Acetic acid | 536 [3.26] | n.d. [<0.01] | 12.6 | 0.00 |
| Formic acid | 538 [3.37] | 549 [0.46] | – | −2.22 |
| Hydrofluoric acid (46%) | 537 [2.35] | 546 [0.60] | 15 ± 2 | −2.94 |
| Trifluoroacetic acid | 539 [3.58] | 550 [0.81] | 3.45 | −3.03 |
| Phosphoric acid (85%) | 540 [2.67] | 549 [0.79] | – | −4.28 |
| Tetrafluoroboric acid (42%) | 536 [––][b] | 547 [0.56] | – | −4.15 |
| Hydrochloric acid (36%) | 539 [2.80] | 550 [0.59] | 1.8 | −4.35 |
| Hexafluorophosphoric acid (55%) | 537 [2.98] | 546 [0.61] | – | – |
| Methanesulfonic acid | 539 [3.43] | 553 [0.75] | 1.6 | −7.86 |
| Sulfuric acid (97%) | 514 [4.28] | 526 [0.85][c] | – | −12.0 |
| Chlorosulfonic acid | 511 [8.11] | 522 [0.68][d] | – | −13.8 |
| Fluorosulfonic acid | 502 [6.01] | 512 [0.90][e] | – | −15.1 |

*n.d.* not detected.

$\lambda_{abs,max}$ and $\lambda_{flu,max}$ denote the maximum absorption and fluorescence emission wavelength (nm), respectively. $\Phi_F$ represents the fluorescence quantum yield.

[a] Excitation wavelength: 510 nm.

[b] Due to low solubility, $\varepsilon$ could not be determined.

[c] Excitation wavelength: 480 nm.

[d] Excitation wavelength: 490 nm.

[e] Excitation wavelength: 470 nm. Measured 12 h after the preparation of solution.

(Supplementary Figs. 30, 31). Under these conditions, most reported analogs, including **12**, lost their emission, whereas B−CN analog **13** still exhibited detectable fluorescence. Therefore, we further evaluated the stability of **11**–**13** under more strongly acidic conditions. When fluorescence spectra were recorded immediately after dissolution in TFA/ H₂O (95/5, v/v) at 10 μM concentration, BODIPYs **11**–**13** exhibited visible emission in the 500–650 nm region. However, the fluorescence intensity of **12** gradually decreased and was almost completely quenched within 1 h. Compound **13** also underwent slight degradation during the same period, with an approximate 15% decrease in fluorescence intensity. In the more acidic medium MSA/H₂O (95/5, v/v), the fluorescence emission of **13** was completely quenched within 15 min at room temperature. In contrast, macrocyclic BODIPY **11** exhibited neither fluorescence quenching nor any change in spectral shape, highlighting its remarkable acid stability relative to that of previously reported BODIPYs. When these BODIPYs **11**–**13** were heated at 180 °C for 48 h in *o*-dichloromethane, the B−CN analog **13** underwent remarkable thermal decomposition, leading to a 71% loss of absorption intensity at 504 nm (Supplementary Fig. 28). Compound **12** also exhibited a 14% decrease in absorbance, whereas the absorption spectrum of cyclic BODIPY **11** remained unchanged after heating. Continuous excitation in *o*-dichlorobenzene at 502 nm with an LED light (29.4 mW mm⁻²) for 2 h caused 66% and 24% loss of absorption intensity for **12** and **13**, respectively, whereas the spectrum of **11** remained virtually unchanged, demonstrating notable photostability (Supplementary Fig. 29).

**Peripheral functionalization and axial ligand exchange**

Acid-stable BODIPY **11** is amenable to both peripheral and axial functionalization, offering tunability of emission wavelength and solubility. When **11** was treated with an excess of bromine, all the pyrrole *β*-

positions were brominated to afford **14** in 98% yield. Single-crystal X-ray diffraction analysis confirmed that the *meso*-CH group remained intact, while it engaged in C-H•••F hydrogen-bonding interactions owing to the increased acidity by the inductive effects of the bromine atoms (Supplementary Fig. 15). Notably, the bowl-depth[33], defined as the distance from the boron atom to the mean plane of the six pyrrole-*β* carbon atoms, was shallower in **14** (1.14 Å) compared to **10** and **11**, which exhibited greater depth of 1.58 and 1.49 Å, respectively (Supplementary Fig. 16). Fluorescence from compound **14** was remarkably attenuated, exhibiting a quantum yield of 1.4% in H₂SO₄, which can be attributed to the internal heavy-atom effect. Instead, photophysical processes associated with the triplet excited state, arising from intersystem crossing, become prominent. In a degassed 2-methyltetrahydrofuran solution containing 5 v/v% MSA, phosphorescence emission was observed at 727 nm upon freezing the sample at 100 K, exhibiting a multiexponential decay with lifetimes of 1.7 and 5.9 ms, whereas no emission was detected at room temperature. Furthermore, irradiation of **14** at 510 nm in the presence of 1,3-diphenylisobenzofuran (DPBF) resulted in a gradual decrease in the characteristic absorption band of DPBF. Comparative analysis using 5,10,15,20-tetraphenylporphyrin as a reference photosensitizer gave a singlet oxygen generation quantum yield of 27% for **14**.

Peripherally aryl-substituted analogs **15**–**19** were prepared using several synthetic strategies. *β*-Hexaphenylated analog **15** was obtained by Suzuki–Miyaura cross-coupling of **14**, whereas *meso*-phenyl-substituted **16** was synthesized by condensation of **9** with trimethyl orthobenzoate in the presence of benzoic anhydride to give **16** in 4% yield. Although the synthetic yields were modest (~3%), the use of acryl chlorides enabled the introduction of 4-methoxyphenyl, 4-cyanophenyl, and 5-methylthiophene-2-yl groups at the *meso*-position of the BODIPY core, affording **17, 18,** and **19,**

**Table 2 | Photophysical properties of peripherally aryl-substituted macrocyclic BODIPYs in various neutral solvents and acidic media**

| Solvent/acid | 15 | 16 | 17 | 18 | 19 |
|---|---|---|---|---|---|
| Dichloromethane | 564/n.d./n.d. | 538/n.d./n.d. | 535/n.d./n.d. | 545/n.d./n.d. | 549/n.d./n.d. |
| Acetonitrile | 559/n.d./n.d. | 534/n.d./n.d. | 532/n.d./n.d. | 541/n.d./n.d. | 545/n.d./n.d. |
| Tetrahydrofuran | 563/n.d./n.d. | 538/n.d./n.d. | 535/n.d./n.d. | 543/n.d./n.d. | 549/n.d./n.d. |
| Dimethyl sulfoxide | 563/n.d./n.d. | 539/n.d./n.d. | 536/n.d./n.d. | 545/n.d./n.d. | 550/n.d./n.d. |
| Ethanol | 562/n.d./n.d. | 536/n.d./n.d. | 533/n.d./n.d. | 543/n.d./n.d. | 548/n.d./n.d. |
| Trifluoroethanol | –[a] | 533/n.d./n.d. | 531/n.d./n.d. | 541/n.d./n.d. | 543/n.d./n.d. |
| Acetic acid | 560/n.d./n.d. | 536/n.d./n.d. | 533/n.d./n.d. | 542/n.d./n.d. | 547/n.d./n.d. |
| Formic acid | 563/593/0.01 | 535/558/0.02 | 531/557/0.01 | 543/569/0.01 | 544/631/0.03 |
| Trifluoroacetic acid | 563/593/0.09 | 532/550/0.01 | 529/550/0.01 | 542/562/<0.01 | 539/622/0.06 |
| Hydrochloric acid (36%) | –[a] | 536/561/0.02 | 534/570/0.01 | 546/578/<0.01 | 546/634/0.01 |
| Methanesulfonic acid | 568/595/0.09 [591/0.37] | 536/560/0.01 [554/0.79] | 533/556/0.04 [551/0.85] | 545/572/<0.01 [570/0.80] | 544/632/0.05 [590/0.34] |
| Sulfuric acid (97%) | 574/595/0.47 | 505/524[a]/0.15[b] | 507/531/0.06[b] | 519/543/0.06[b] | 522/605/0.09[b] |
| Chlorosulfonic acid | –[d] | 503/522[a]/0.14[b] | 508/529/0.02[b] | 521/539/0.01[b] | 520/608/0.13[c] |
| Fluorosulfonic acid | –[d] | –[d] | 501/526/0.01[b] | 513/535/<0.01[b] | 511/604/0.07[b] |

Values in each cell are given as $\lambda_{abs,max}$ (nm)/$\lambda_{flu,max}$ (nm)/$\Phi_F$. Unless otherwise noted, fluorescence spectra were recorded with an excitation wavelength of 510 nm. Values in brackets indicate the fluorescence maximum $\lambda_{flu,max}$ (nm) and quantum yield $\Phi_F$ measured in frozen methanesulfonic acid solutions.

*n.d.* not detected.

[a] Compound was not sufficiently soluble.

[b] Excitation wavelength: 480 nm.

[c] Excitation wavelength: 500 nm.

[d] Multi-protonation at the BODIPY core occurred under these conditions, rendering direct comparison of photophysical data inappropriate.

respectively. The peripherally aryl-substituted derivatives **15–19** also exhibited protonation-induced turn-on fluorescence behavior (Table 2 and Supplementary Figs. 33–37). While these compounds were non-fluorescent in neutral organic solvents, including tetrahydrofuran, acetonitrile, and ethanol, distinct substituent effects on fluorescence emission were observed in acidic media. Figure 5c shows the absorption and fluorescence spectra of **15–19** recorded in 97% sulfuric acid. Among this series, β-hexaphenyl analog **15** exhibited a distinct red-shift of the absorption band (574 nm), while the lowest energy bands of other analogs **11** and **16–19** remained in the range of 505–522 nm. Consistent with the absorption behavior, the fluorescence emission of **15** was also red-shifted to 595 nm, with a fluorescence quantum yield of 47%. The thienyl-substituted analog **19** exhibited a pronounced red shift of the emission spectrum to 605 nm with a large Stokes shift, accompanied by spectral broadening and a low-energy tail extending beyond 750 nm. This behavior can be attributed to intramolecular charge-transfer interactions between the thienyl moiety and the BODIPY core[42]. Consistent with this interpretation, a relatively large slope was observed for **19** in the Lippert–Mataga plot analysis compared with analogs **15–18** (Supplementary Fig. 38). While the substituent effects at the 4-position of the *meso*-phenyl ring on absorption and emission wavelength were modest, the fluorescence quantum yields of *meso*-substituted analogs **16–19** decreased considerably. Such *meso*-aryl substituent effects have been reported for conventional BODIPYs, in which rotation of the aryl groups in the excited state contributes to non-radiative decay pathways[43,44]. Table 2 shows that fluorescence quantum yields tend to increase in more viscous acids, such as sulfuric acid, compared with TFA and formic acid. Based on this trend, we measured fluorescence quantum yields of **15–19** in frozen MSA solutions and found that the values increased markedly to 0.37, 0.79, 0.85, 0.80, and 0.34, respectively, compared with those measured in liquid solution (0.09, 0.01, 0.04, <0.01, and 0.05). These observations are attributable to the restricted aryl ring rotation in frozen MSA, which suppresses non-radiative decay pathways. Taken together, these results demonstrated that macrocyclic BODIPYs allow

tuning of emission wavelength and fluorescence quantum yields by substituent effects of peripheral aryl groups without deboronation in highly acidic environments.

Axial ligand exchange on the boron atom offers a facile, orthogonal functionalization strategy for macrocyclic BODIPYs, enabling the modulation of solubility without altering the π-electronic structure of the BODIPY core. Axially methoxy-substituted analog **10** underwent ligand exchange reaction with carboxylic acids in a manner similar to subporphyrins[45]. Azeotropic removal of methanol in the presence of tridecafluorooctanoic acid in chloroform afforded axially fluorous-tagged derivative **20** in 86% yield. Although macrocyclic BODIPYs **10** and **11** were insoluble in fluorous solvents, **20** was sufficiently soluble in perfluoromethylcyclohexane (49 mg/L), enabling fluorescence switching in the fluorous phase in response to acids (Fig. 5). We examined perfluorooctanoic acid (PFOA), which is a per- and poly-fluoroalkyl substance (PFAS)[46,47], as a fluorous-soluble acid to serve as a fluorescence trigger. Compound **20** was non-fluorescent in perfluoromethylcyclohexane. Upon addition of PFOA to a 72 μM solution of **20**, visible fluorescence from in situ-generated **20**•H⁺ became observable at approximately 1 mM. Although the detection sensitivity of **20** was modest compared with that of known PFOA sensors, this result provides a potential scaffold for the development of fluorescent sensors in fluorous phases.

**Potential applications**

Superacid-resistant BODIPY **11** serves as a promising fluorescent indicator for staining strongly acidic materials. Nafion is a perfluoroalkylsulfonic acid-based polymer widely used as a solid polymer electrolyte and acid catalyst, but its fluorescent staining with conventional BODIPYs is often unsuccessful owing to its strong acidity. When Nafion beads (φ: 4.0 mm) were immersed in a 0.25 mM toluene solution of **11** for 2 days, the beads changed from colorless to orange, and intense reddish-orange fluorescence arising from **11**•H⁺ was observed (Fig. 6). Under the same staining conditions, conventional BODIPY **12** resulted in only a slight color change, but its fluorescence emission was barely detectable due to deboronation. The **11**-stained Nafion beads

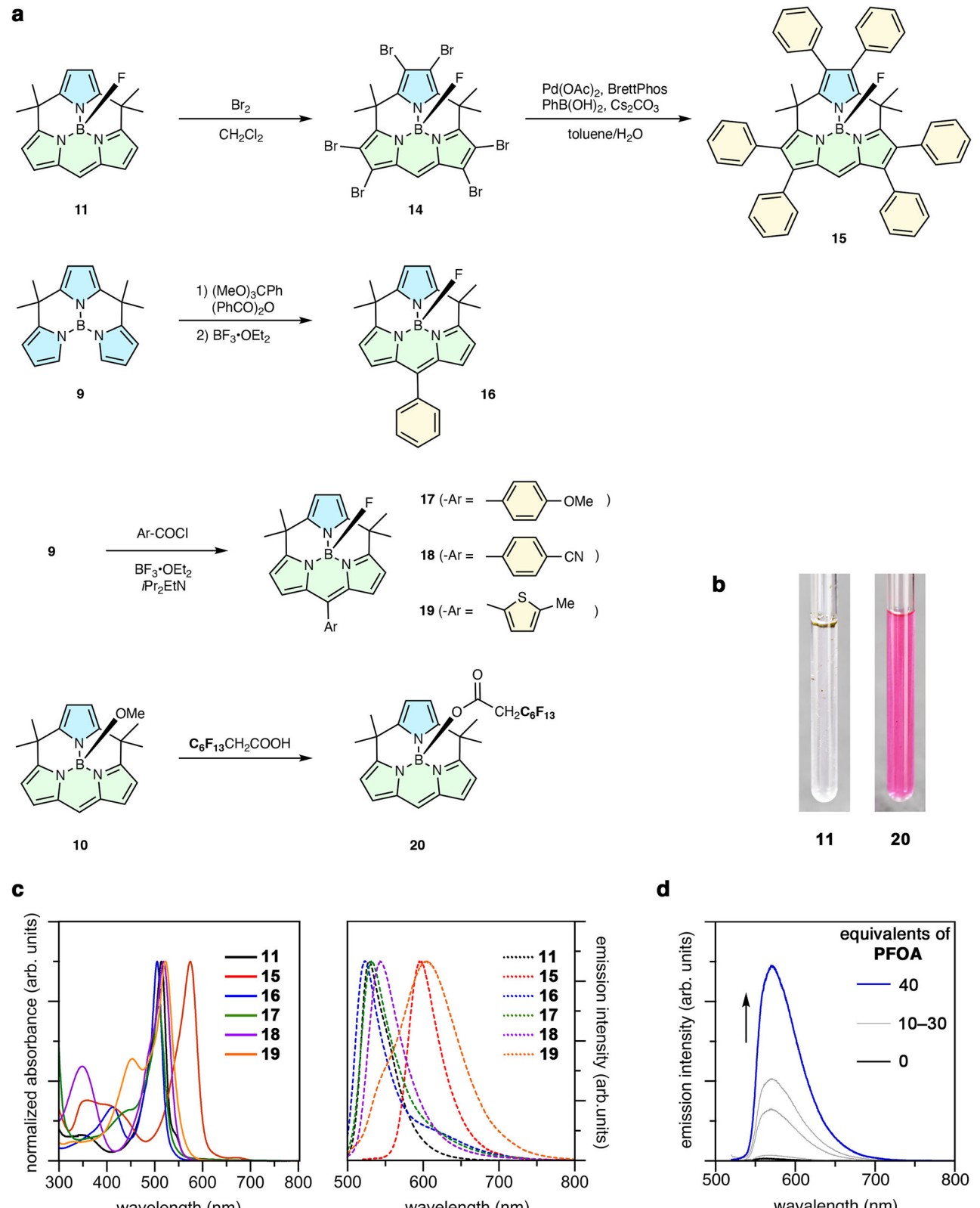

**Fig. 5 | Peripheral and axial functionalization of macrocyclic BODIPYs.**
**a** Synthetic schemes for β-hexaphenyl, *meso*-aryl and axially perfluoroalkyl-substituted analogs **15**–**20**. **b** Photographs comparing the solubility of compounds **11** and **20** in perfluoromethylcyclohexane (0.5 mg/500 μL); **11** formed a suspension and **17** yielded a clear solution. **c** UV-vis absorption (solid lines) and fluorescence spectra (dotted lines; excited at 480 nm, except for **15** at 510 nm) of **11, 15, 16, 17, 18**, and **19** in 97% sulfuric acid. **d** Fluorescence spectra of **20** (72 μM; excited at 510 nm) in perfluoromethylcyclohexane upon addition of PFOA (0–40 equivalents).

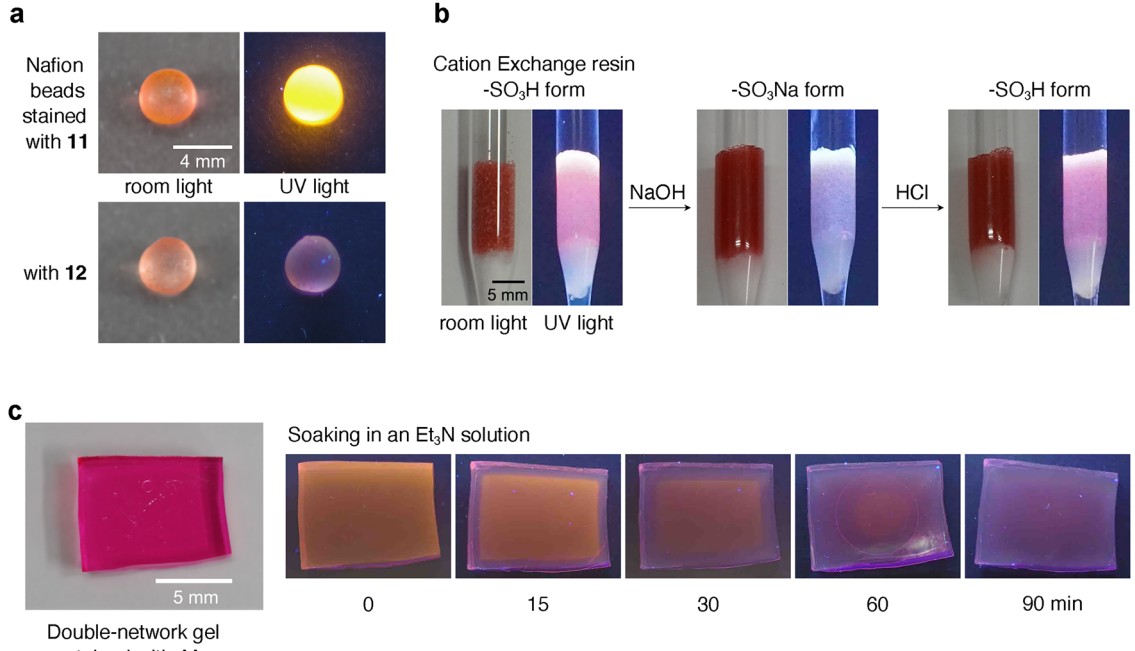

**Fig. 6 | Staining of strongly acidic materials with superacid-resistant BOD-IPY 11. a** Comparison of the fluorescent staining of Nafion beads using **11** and **12**. **b** Fluorescence switching of **11** immobilized on a sulfonyl-functionalized cation exchange resin. **c** Sulfonylated double network gel stained with **11** under ambient light (left) and time-dependent fluorescence attenuation under UV light (365 nm) upon soaking in a 0.1 M ethanol solution of triethylamine (right).

remained fluorescent for more than 1 week without significant loss of intensity, whereas exposure to triethylamine vapor quenched the emission due to the deprotonation of **11·H+** to neutral **11**.

BODIPY **11** also functions as an acid indicator for sulfonyl-functionalized cation exchange resins. When the resin in its -SO₃H form was soaked in an acetonitrile solution of **11**, it turned reddish-orange, indicating the immobilization of **11**. A leaching test, performed by passing distilled water through a column packed with the **11**-stained resin, confirmed that **11** was firmly retained, because there was no detectable leaching. In the -SO₃H form, the **11**-stained resin emitted fluorescence under UV irradiation owing to in situ-generated **11·H+**. The fluorescence disappeared when a 1.0 M aqueous sodium hydroxide solution was passed through, converting the resin into its -SO₃Na form. Subsequent treatment with 1.0 M hydrochloric acid restored the fluorescence without detectable leaching of **11**, demonstrating its reversible acid-indicating capacity. Notably, compound **11** also remained stable under basic conditions and did not undergo deboronation.

The interconversion between fluorescent **11·H+** and non-fluorescent **11** was utilized to visualize the penetration of a contacting solution into acidic gel materials. Double network gels are widely recognized for their versatile applications, such as self-growing material[48] and artificial cartilage[49], and also include strongly acidic sulfonylated analogs[50]. A sulfonated double network organogel (approximately 20 mm × 15 mm × 5 mm) was successfully stained by immersing it in a 0.25 mM ethanol solution of **11** for 12 h. Under UV light, the stained gel emitted orange fluorescence uniformly throughout the gel. When the stained gel was soaked in a 0.1 M ethanol solution of triethylamine at room temperature, the fluorescence was gradually quenched from its edges, and completely disappeared within 90 min. This enabled clear visualization of the penetration process of the base into the organogel. The results described above demonstrate that staining with a superacid-resistant macrocyclic BODIPYs complements conventional BODIPY-based staining by enabling bright emission and reversible acid–base switching in strongly acidic materials. This broadens the scope of fluorescent detection and sensing with BODIPYs.

## Discussion

In this study, we discovered a synergistic boron-chelation effect in globally nonaromatic tripyrrolic macrocycles that enables reversible protonation via formation of a tetracoordinated boronium cation without deboronation. Unlike SubPcs and SubPors, which retain boron chelation through the formation of tricoordinated borenium cations, this system undergoes dearomative protonation at a π-conjugationally isolated pyrrole unit under acidic conditions, resulting in only minor perturbations to the boron coordination environment and π-electronic conjugation of the remaining two pyrrole units. Therefore, the BODIPY core embedded within the tripyrrolic macrocycle retains its intrinsic photophysical properties, including strong visible absorption and fluorescence emission with small Stokes shifts, even under strongly acidic conditions. This insight establishes a general and rational design principle for constructing superacid-resistant BODIPYs based on boron–tripyrrolic macrocycle synergy. The superacid-resistant BODIPYs retained bright fluorescence emission with quantum yields of up to 90% in non-diluted superacids, without undergoing deboronation. In addition to their exceptional acid stability, the macrocyclic BODIPYs demonstrated remarkable thermal and photo-stability compared with conventional BODIPYs. Furthermore, peripheral substitution with aryl groups at the pyrrole-β or *meso*-positions enabled the modulation of the emission wavelengths through intramolecular charge-transfer interactions and aryl-ring rotation, even in highly acidic environments, whereas boron-axial ligand exchange allowed facile control of solubility in fluorous solvents. Taking advantage of their protonation-induced turn-on fluorescence, we used these dyes as fluorescent acid indicators of strongly acidic materials such as Nafion and sulfonylated gels, which are otherwise difficult to stain using conventional BODIPYs. By combining exceptional acid resistance with the intrinsic photophysical advantages of BODIPYs, this work extends the applicability of BODIPYs to non-aqueous superacidic media. In such environments, the absence of the leveling effect of water typically prevents conventional BODIPYs from retaining their fluorescence. This capability opens a broad spectrum of future applications, including the fluorescent labeling of

strongly acidic polymers and catalysts, the staining of zeolite-type minerals, and the visualization of acidophilic microorganisms in extreme environments. Such applications underscore the potential of superacid-resistant BODIPYs to extend the frontier of fluorescence imaging and sensing into domains where traditional BODIPYs have been left largely unexplored.

## Methods

Solvents and reagents were purchased from WAKO Pure Chemical Industries Ltd., TCI Co., Ltd., Kanto Chemical Co., Inc., or Sigma-Aldrich Co., and were used without further purification unless otherwise mentioned. Boron(III)-calix[3]pyrrole complex **1**[26] and tripyrrane **8**[51] were prepared according to a reported procedure. All $^1$H and $^{13}$C NMR spectra were recorded using JEOL JMN-ECS400 or JMN-ECZ600R spectrometers. Chemical shifts were reported in ppm relative to the internal standard tetramethylsilane ($\delta = 0.00$ ppm for $^1$H NMR in CDCl$_3$), the external standard boron trifluoride ethyl etherate ($\delta = 0.00$ ppm for $^{11}$B NMR in CDCl$_3$), the external standard hexafluorobenzene (–162.9 ppm for $^{19}$F NMR in CDCl$_3$), or a solvent residual peak ($\delta = 77.16$ ppm for $^{13}$C NMR in CDCl$_3$). Thin-layer chromatography was performed on a silica gel sheet, MERCK silica gel 60 F254. Preparative scale separations were performed by means of gravity column chromatography over silica gel (Wakosil® 60. 64–210 μm). Infrared spectra were measured using a JASCO Co. FT/IR-4600 spectrometer. ESI-TOF-MS spectra were recorded on a Thermo Scientific Executive spectrometer. Elemental analyses were carried out using an Exceter Analytical, Inc. CE440 or MICRO CORDER JM10. UV/Vis absorption spectra were recorded on a JASCO V-770, V-670 or SHIMADZU UV-1800 spectrophotometers. Fluorescence spectra were recorded on HITACHI F-7000 or JASCO FP-8550 spectrometers. Fluorescence lifetimes were measured using an Edinburgh FLS1000 photoluminescence spectrometer. Fluorescence quantum yields were measured using a Hamamatsu Photonics Quantaurus QY C11347-01 system or an Edinburgh FLS 1000 photoluminescence spectrometer. Phosphorescence spectra and decay curves were recorded using a JASCO FP-8550 spectrometer with a cryostat (Oxford Instruments, OptistatDN) and a temperature controller (Oxford Instruments, ITC502S). Single crystal X-ray diffraction data were obtained using a Rigaku XtaLAB P200 diffractometer equipped with a PILATUS200K detector, which uses a multilayer mirror (MoK$_\alpha$ radiation $\lambda = 0.71073$ Å) or a Rigaku XtaLAB Synergy-R/DW instrument equipped with a HyPix-6000HE detector, which uses a monochromated mirror. Cyclic voltammograms were measured with an ALS Model 660E electrochemical analyzer using a three-electrode system.

## Data availability

The X-ray crystallographic coordinates for structures reported in this study have been deposited at the Cambridge Crystallographic Data Centre (CCDC), under deposition numbers 2482056 (**2**), 2482057 (**10**), 2482058 (**11**), and 2482059 (**14**). These data can be obtained free of charge from the Cambridge Crystallographic Data Centre via www.ccdc.cam.ac.uk/data_request/cif. All data generated or analyzed during this study are included in this published article and its Supplementary information. All data are available from the corresponding author upon request. Coordinate files are provided with this paper. Source data are provided with this paper.

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

## Acknowledgements

This work was partly supported by a JSPS Grant-in-Aid for Challenging Research (Exploratory) (No. JP24K2178704), a JST FOREST Program (No. JPMJFR211H) and the Asahi Glass Foundation, of which Y. Inokuma is the principal investigator. This work was supported by a JSPS grant-in-aid for Early-Career Scientists (grant no. 25K18056) to Y. Ide, and the NorthTec Foundation and grant-in-aid for scientific research (grant no. 25K08625) to T.Y. The authors thank Mr. Taichi Sano for his support in the synthetic experiments. The Institute for Chemical Reaction Design and Discovery (ICReDD) was established by World Premier International Research Initiative (WPI), MEXT, Japan.

## Author contributions

Y. Inokuma designed the study, supervised the project, and wrote the first version of the draft. K.W., G.H., Y. Inaba, Y.T., S.M., Y. Ide, and T.Y. performed syntheses and measurements for cyclic BODIPYs. T.N. and J.P.G. worked on the synthesis of sulfonated gels. M.G. carried out quantum chemical calculations. Y. Y., Y. K., and Y. H. performed photophysical analyses, including phosphorescence emission and decay measurements, as well as singlet oxygen generation measurements.

## Competing interests

The authors declare no competing interests.
