## [Transparent Peer Review file · Nature Communications]

Superacid-resistant macrocyclic BODIPYs

Corresponding Author: Professor Yasuhide Inokuma

Version 0:

Reviewer comments:

Reviewer #1

(Remarks to the Author)

This work presents a class of macrocyclic BODIPYs that utilize a tripyrrolic macrocycle for synergistic coordination to the boron center, with a focus on maintaining coordination without deboronation in superacidic environments and achieving protonation-triggered luminescence switching. The authors report high quantum yields and long-term stability in superacids such as fluorosulfonic acid, and they demonstrate application scenarios including the staining of Nafion and sulfonated gels. The overall novelty and visibility are good. However, for submission to Nature Communications, the mechanistic depth and the persuasiveness of the applications need to be strengthened. My major concern is that the overall synthetic strategy shows significant limitations in its generality, restricting access to key classes of BODIPY derivatives such as meso-aryl substituted and aza-BODIPY analogues. The manuscript's text could also be further condensed and more organized. Thus, major revisions are recommended before reconsideration. Specific recommendations and comments that would improve the manuscript are listed below:

1. The manuscript attributes the fluorescence quenching of compound 11 to a PET mechanism, supported solely by TD-DFT calculations. While the calculated orbital distributions are consistent with this proposal, experimental validation is crucial for such a central mechanistic claim. I recommend that the authors provide experimental evidence to substantiate the PET process. Techniques such as cyclic voltammetry should be used to obtain donor and acceptor redox potentials, then estimate $\Delta G(\text{PET})$ with the Rehm–Weller approximation for the neutral and protonated states.
2. Hexabrominated 14 is reported to be non-fluorescent. Please examine phosphorescence and quantify singlet oxygen generation. The internal heavy atom effect from six bromine substituents should enhance intersystem crossing and populate a long-lived triplet, which could make 14a useful photosensitizer.
3. The fluorescence spectrum of 15 in concentrated H₂SO₄ (in Figure 5c) clearly shows a complex, broad emission band with a distinct dual-emissive character (peaks at approx. 575 nm and 600 nm). The current description as merely "red-shifted" is insufficient and overlooks this important feature.
4. The hexaphenyl derivative 15, with its six rotatable peripheral phenyl groups, is a prime candidate for exhibiting AIE. The rotation of these phenyls can serve as a non-radiative decay channel in solution, which would be blocked upon aggregation. The fluorescence observed in highly viscous sulfuric acid (QY = 0.16) may already hint at this restriction of intramolecular motion.

Minor points:

1. To significantly improve the clarity and accessibility of the results, I strongly recommend that the authors compile the spectroscopic and photophysical data for all new compounds into a summary table.
2. Supplementary Table 2 should be included in the TEXT or shown as a plot.

Reviewer #2

(Remarks to the Author)

I co-reviewed this manuscript with one of the reviewers who provided the listed reports. This is part of the Nature

Communications initiative to facilitate training in peer review and to provide appropriate recognition for Early Career Researchers who co-review manuscripts.

Reviewer #3

(Remarks to the Author)

The manuscript describes the synthesis and characterization of a new family of contracted pyrrolic macrocycles that structurally resemble macrocyclic BODIPY analogues with a tridentate cavity. The authors show that these frameworks can tightly coordinate boron, imparting exceptional acid resistance — a feature of high relevance since BODIPY dyes are notoriously prone to deborylation under acidic conditions. The reported systems are stable even at $\text{pH} \approx 1$ and can be post-functionalized at both the meso and peripheral positions, allowing for subsequent derivatization. The photophysical behavior is explored and potential applications are demonstrated in polymer staining, pH sensing, and imaging in acidic media. The work is well executed, and the data are of high experimental quality.

However, I do not believe the manuscript meets the novelty and impact threshold required for Nature Communications. The study, while solid, represents a specialized development within the already well-established field of boron–pyrrole chromophores rather than a conceptual or technological breakthrough of broad interest. I would therefore not recommend publication in its current form. Other more chemistry specialized journals are recommended instead.

Major Comments

1. Limited novelty relative to existing acid-resistant fluorophores: The claim of exceptional acid stability is not unique. Several acid-tolerant fluorescent probes and chromophores have been previously reported (e.g., *Anal. Chim. Acta* 2014, 816, 1; *Chem. Soc. Rev.* 2014, 43, 4629; *Biosens. Bioelectron.* 2024, 116638). The authors should position their compounds explicitly against these precedents and demonstrate in quantitative terms why their system is superior or conceptually distinct. Without this comparison, the contribution appears incremental.
2. Inconsistencies in the mechanistic rationale and background: e.g., the introduction states that “acid accelerates axial exchange in subporphyrinoids.” This is not generally the case; protonation at meso positions or ring opening are far more common acid-induced processes (see *New J. Chem.* 2018, 42, 10478). Clarification is needed, as this assertion currently undermines the mechanistic logic.
3. Redundancy with previous work on boron–tridentate macrocycles: The notion of embedding boron within a tridentate pyrrolic environment lacks strong novelty, particularly given the authors' own prior studies on C_3P frameworks and the established chemical robustness of subporphyrinoids. The manuscript would benefit from a clearer statement of what genuinely new coordination or photophysical behavior arises here beyond what is already known for SubPcs, SubPors, or related contracted systems.
4. The proposed “acid-induced fluorescence switching” mechanism—arising from suppression of intramolecular quenching—is a recurring concept in chromophore design and has been widely documented. The authors should better contextualize their findings within this literature and specify whether any mechanistic feature here is truly unprecedented.
5. Limited optical tunability and modest spectral range: the emission remains confined to the 500–600 nm region. Extension into the near-infrared, which is the current frontier in bioimaging, would represent genuine novelty and impact. Similarly, the observed red-shifts upon peripheral aryl substitution are small (compared with subporphyrinoids or subporphyrazines (e.g., *J. Org. Chem.* 2020, 85, 1152)), and the substantial decrease in quantum yield upon functionalization raises questions about the practical value of these derivatives.

Additional Remarks

1. Literature coverage

Several seminal works are missing. For instance, the comprehensive *Chem. Rev.* 2007, 107, 4891–4932 on BODIPYs should be cited, as well as the most recent reviews on subporphyrinoids (*Chem. Soc. Rev.* 2022, 51, 9482; *Trends Chem.* 2023, 5, 279). The current bibliography underrepresents key prior art, weakening the contextual framing.

2. Line 227: “solubile” → “soluble”. Please also perform a careful language and consistency revision

In summary, the manuscript presents technically competent work and interesting synthetic chemistry, but it falls short of the conceptual advance and broad significance expected for Nature Communications. The acid stability and post-functionalization are incremental improvements within an already mature class of compounds. A more comprehensive comparison with known acid-resistant fluorophores, clarification of the mechanistic novelty, and demonstration of distinctive optical performance (e.g., NIR emission, tunability) would be required for reconsideration.

Reviewer #4

(Remarks to the Author)

This is a very interesting paper that describes the synthesis, characterization and application of superacid resistant cyclic BODIPYs. I believe the work presented is significant to the field of BODIPYs and their practical applications; a clear comparison between the new reported acid-stable cyclic BODIPYs and non-cyclic BODIPYs is investigated by the authors. All compounds are adequately characterized by NMR and HRMS. The methodologies used for the characterization of the neutral and positively charged species are all adequate and the conclusions are supported by the results obtained. The Figures are adequate and informative. I suggest that the authors correct a few typos in the manuscript, such as in lines 227 and 248. In addition, the Discussion section is more of a Conclusions section.

Version 1:

Reviewer comments:

Reviewer #1

(Remarks to the Author)

The authors have appropriately addressed the major concerns. However, to ensure reproducibility, the methodological details in the Supporting Information needs to be expanded in detail prior to acceptance. The newly conducted experimental procedures, such as phosphorescence and singlet oxygen measurements, were wrapped in the captions of Figures. They should have a more detailed experimental section. Other than that, this excellent manuscript is recommended for acceptance.

Reviewer #2

(Remarks to the Author)

Reviewer #3

(Remarks to the Author)

The authors have clarified all the concerns raised by this referee. With these changes, this referee is now convinced that the work is suitable for Nat. Comm, and therefore recommends its publication.

Reviewer #4

(Remarks to the Author)

[Editor's Note: This reviewer left remarks to the editor indicating all technical concerns have been addressed]

Point-by-Point Response to the Reviewers' comments

(*blue*: original comments by reviewers, *black*: our response)

Reviewer 1

This work presents a class of macrocyclic BODIPYs that utilize a tripyrrolic macrocycle for synergistic coordination to the boron center, with a focus on maintaining coordination without deboronation in superacidic environments and achieving protonation-triggered luminescence switching. The authors report high quantum yields and long-term stability in superacids such as fluorosulfonic acid, and they demonstrate application scenarios including the staining of Nafion and sulfonated gels. The overall novelty and visibility are good. However, for submission to Nature Communications, the mechanistic depth and the persuasiveness of the applications need to be strengthened. My major concern is that the overall synthetic strategy shows significant limitations in its generality, restricting access to key classes of BODIPY derivatives such as meso-aryl substituted and aza-BODIPY analogues. The manuscript's text could also be further condensed and more organized. Thus, major revisions are recommended before reconsideration. Specific recommendations and comments that would improve the manuscript are listed below:

We thank the reviewer for the careful evaluation and for recognizing the novelty and relevance of our study, as well as for providing constructive comments that helped us improve the mechanistic discussion and overall clarity. In response to the reviewer's major concern regarding the synthetic generality of macrocyclic BODIPYs, we have expanded the scope to include additional *meso*-aryl derivatives, which were successfully synthesized using the corresponding acyl chlorides. The successful use of acyl chlorides represents a notable synthetic advance, enabling broad access to *meso*-aryl-substituted derivatives of this macrocyclic BODIPY framework, despite the modest yields. Introduction of a *meso*-thienyl moiety resulted in a remarkable red-shift of the emission band owing to intramolecular charge-transfer interactions, a behavior commonly observed in conventional BODIPYs and supported by Lippert–Mataga plots. To summarize the effects of peripheral substitution, the photophysical data of peripherally aryl-substituted analogues **15–19** are shown in Table 2, enabling direct comparison with the detailed data for the parent compound **11** summarized in Table 1. Although attempts to synthesize aza-BODIPY analogues using precursor **9** were unsuccessful at this stage, we believe that the demonstrated expansion to *meso*-aryl derivatives effectively addresses the reviewer's concern regarding synthetic scope and wavelength tunability.

Our detailed responses to the other comments are given below.

1. The manuscript attributes the fluorescence quenching of compound 11 to a PET mechanism, supported solely by TD-DFT calculations. While the calculated orbital distributions are consistent with this proposal, experimental validation is crucial for such a central mechanistic claim. I recommend that the authors provide experimental evidence to substantiate the PET process. Techniques such as cyclic voltammetry should be used to obtain donor and acceptor redox potentials, then estimate $\Delta G(\text{PET})$ with the Rehm–Weller approximation for the neutral and protonated states.

The free energy change for the electron transfer process ($\Delta G(\text{PET})$) in BODIPY **11** under neutral conditions, estimated using the Rehm–Weller equation, was -0.46 eV, indicating that the PET process is thermodynamically favorable (Supplementary Fig. 26). While the $\Delta G(\text{PET})$ value was obtained based on the 1st oxidation (0.70 V) and reduction (-1.53 V) potentials of **11** measured by cyclic voltammetry in CH_2Cl_2 , the addition of 5 v/v% TFA resulted in near-complete attenuation of the 1st oxidation peak, presumably due to the protonation at the pyrrole moiety. Consequently, the apparent $\Delta G(\text{PET})$ estimated under the acidic conditions became positive, indicating suppression of the PET process. These results are consistent with the DFT and spectroscopic analyses, thereby supporting the proposed PET mechanism. We have added these results and discussions in the main text.

2. Hexabrominated 14 is reported to be non-fluorescent. Please examine phosphorescence and quantify singlet oxygen generation. The internal heavy atom effect from six bromine substituents should enhance intersystem crossing and populate a long-lived triplet, which could make 14a useful photosensitizer.

Phosphorescence from brominated BODIPY **14** was observed at 727 nm in a frozen 2-methyltetrahydrofuran matrix containing 5 v/v% MSA at 100 K under degassed conditions. The phosphorescence decay of **14** was fitted with a multiexponential function, giving lifetimes of 1.7 and 5.9 msec (Supplementary Fig. 32). Although we initially described compound **14** as non-fluorescent, careful analysis of the fluorescence spectrum in the same solvent revealed that its fluorescence quantum yield, while low, was measurable and determined to be 1.4% at room temperature.

Singlet oxygen generation efficiency was also evaluated by monitoring the photooxidation of 1,3-diphenylisobenzofuran (DPBF), using 5,10,15,20-tetraphenylporphyrin as a standard photosensitizer and was determined to be 27%.

3. The fluorescence spectrum of 15 in concentrated H_2SO_4 (in Figure 5c) clearly shows a complex, broad emission band with a distinct dual-emissive character (peaks at approx. 575 nm and 600 nm). The current description as merely "red-shifted" is insufficient and overlooks this important feature.

4. The hexaphenyl derivative 15, with its six rotatable peripheral phenyl groups, is a prime candidate for exhibiting AIE. The rotation of these phenyls can serve as a non-radiative decay channel in solution, which would be blocked upon aggregation. The fluorescence observed in highly viscous sulfuric acid (QY= 0.16) may already hint at this restriction of intramolecular motion.

Thank you for these important suggestions. We carefully re-examined the fluorescence measurements of **15** and found that, due to its limited solubility in sulfuric acid, the compound was not fully dissolved at the time of previous measurement, resulting in undissolved solid material remaining suspended in the sample solution. The revised spectrum of **15** in sulfuric acid exhibited a single fluorescence emission band at 595 nm with a quantum yield of 0.47,

which appears as a mirror image of the most intense absorption band at 574 nm, consistent with the behavior observed for **11**. Although we analyzed the absorption and fluorescence spectra of **15** in various acidic media (see Table 2 and Supplementary Fig. 33), no clear spectroscopic evidence for aggregation was observed under the conditions examined. Instead, we observed the distinct red-shifts in both the absorption and fluorescence bands compared to **11**.

All peripherally aryl-substituted analogues **15–19** exhibited enhanced fluorescence quantum yields in concentrated sulfuric acid (viscosity at 298 K, $\eta = 24.2$ mPa·s) compared with those in formic acid ($\eta \sim 1.8$) and TFA ($\eta \sim 0.86$). Furthermore, freezing the MSA solutions of **15–19** also led to remarkably increased fluorescence quantum yields. These results indicate that non-radiative decay pathways are suppressed as a result of restricted intramolecular rotation of the aryl groups.

Minor points:

- 1. To significantly improve the clarity and accessibility of the results, I strongly recommend that the authors compile the spectroscopic and photophysical data for all new compounds into a summary table.*
- 2. Supplementary Table 2 should be included in the TEXT or shown as a plot.*

Supplementary Table 2 has been incorporated into the main text as Table 1, with additional information on absorption maxima and molar extinction coefficients. The photophysical properties of the peripherally aryl-substituted analogues, including newly synthesized *meso*-aryl derivatives, are also summarized in Table 2 of the main text to facilitate direct comparison.

Reviewer 2

We thank the reviewer for taking part in the peer-review process and for the valuable time and effort. We appreciate the inclusion of early-career researchers in the review, as such multi-perspective evaluations enhance the rigor of the review process and help us further improve the quality of our work.

Reviewer 3

The manuscript describes the synthesis and characterization of a new family of contracted pyrrolic macrocycles that structurally resemble macrocyclic BODIPY analogues with a tridentate cavity. The authors show that these frameworks

can tightly coordinate boron, imparting exceptional acid resistance — a feature of high relevance since BODIPY dyes are notoriously prone to deborylation under acidic conditions. The reported systems are stable even at $\text{pH} \approx 1$ and can be post-functionalized at both the meso and peripheral positions, allowing for subsequent derivatization. The photophysical behavior is explored and potential applications are demonstrated in polymer staining, pH sensing, and imaging in acidic media. The work is well executed, and the data are of high experimental quality.

However, I do not believe the manuscript meets the novelty and impact threshold required for Nature Communications. The study, while solid, represents a specialized development within the already well-established field of boron–pyrrole chromophores rather than a conceptual or technological breakthrough of broad interest. I would therefore not recommend publication in its current form. Other more chemistry specialized journals are recommended instead.

We appreciate the reviewer's careful assessment of our work and the important concerns raised regarding its novelty and conceptual significance. Although the central advance of this study lies in the discovery of a previously unrecognized protonation behavior in pyrrole–boron complexes bearing a π -conjugationally isolated pyrrole unit, we recognize that our initial manuscript did not sufficiently clarify how this behavior differs mechanistically from the well-studied acid responses of conventional BODIPYs as well as subporphyrins and subphthalocyanines. As a result, the conceptual novelty of the present system may not have been clearly conveyed, as also pointed out in the reviewer's major comments.

The striking difference in the protonation behavior of the tripyrrolic macrocycle in the present work, compared with other tripyrrolic boron macrocycles such as subporphyrins and subphthalocyanines, is the completely reversible, dearomative protonation at the π -conjugationally isolated pyrrole unit. While subporphyrins retain boron chelation within a 14π -aromatic macrocycle via formation of a tricoordinated borenium cation under acidic conditions, globally non-aromatic systems such as calix[3]pyrroles and macrocyclic BODIPYs exhibit a fundamentally different response. In these systems, protonation leads to a tetracoordinated boronium cation, thereby maintaining stable chelation of the boron atom. Importantly, the boronium-forming pathway enabled by a π -conjugationally isolated pyrrole unit represents a distinct and previously unrecognized strategy for achieving superacid resistance in BODIPY chromophores. Although boron–calix[3]pyrrole complex **1** was known to tolerate acidic conditions, the present study clarifies the underlying protonation behavior and demonstrates how this mechanism can be translated into the rational design of superacid-resistant BODIPY dyes.

To clarify this conceptual distinction, and in direct response to the reviewer's major comments, we have revised the Introduction and Discussion sections to more clearly articulate the unique protonation mechanism and its implications for the design of superacid-resistant BODIPY chromophores (please see the yellow-highlighted revisions in the manuscript).

Major Comments

1. Limited novelty relative to existing acid-resistant fluorophores: The claim of exceptional acid stability is not unique.

Several acid-tolerant fluorescent probes and chromophores have been previously reported (e.g., Anal. Chim. Acta 2014, 816, 1; Chem. Soc. Rev. 2014, 43, 4629; Biosens. Bioelectron. 2024, 116638). The authors should position their compounds explicitly against these precedents and demonstrate in quantitative terms why their system is superior or conceptually distinct. Without this comparison, the contribution appears incremental.

This important suggestion prompted us to strengthen the quantitative comparison with previously reported acid-tolerant fluorophores. We added Supplementary Fig. 31, which summarizes representative BODIPY structures reported in the literature and the most acidic pH conditions under which fluorescence emission has been observed for each analogue. As shown in the figure, most BODIPY analogues have been evaluated in aqueous media at pH ≥ 0 . In contrast, our BODIPY **11** remains fluorescent even in 36% hydrochloric acid (pH < -1 ; Supplementary Fig. 30). Among the BODIPY derivatives surveyed in the literature, B–CN analogues have been reported to exhibit particularly high acid tolerance. Consistent with these reports, we also confirmed that compound **13** retained BODIPY-based fluorescence in 36% hydrochloric acid, while conventional BODIPYs such as **12** could not retain fluorescence under these conditions. Therefore, we further investigated the acid stability of these dyes through comparative experiments (Fig. 4c). We have also incorporated this discussion into the main text to clarify the quantitative and conceptual distinction from previously reported systems.

2. Inconsistencies in the mechanistic rationale and background: e.g., the introduction states that “acid accelerates axial exchange in subporphyrinoids.” This is not generally the case; protonation at meso positions or ring opening are far more common acid-induced processes (see New J. Chem. 2018, 42, 10478). Clarification is needed, as this assertion currently undermines the mechanistic logic.

We thank the reviewer for pointing out this important issue. We agree that the acid responses of subphthalocyanines and subporphyrins should not be generalized, and that our original wording was misleading. We have revised the Introduction to clarify that, as follows:

“Under acidic conditions, protonation or Lewis acid coordination at the *meso*-azomethine units is commonly observed for subphthalocyanines, whereas acids often promote boron-axial ligand exchange at the boron center of hydroxo- or alkoxo-substituted subporphyrins.” This statement is now supported by an appropriate literature citation (*New J. Chem.* **2018**, 42, 1622).

3. Redundancy with previous work on boron–tridentate macrocycles: The notion of embedding boron within a tridentate pyrrolic environment lacks strong novelty, particularly given the authors’ own prior studies on C₃P frameworks and the established chemical robustness of subporphyrinoids. The manuscript would benefit from a clearer statement of what genuinely new coordination or photophysical behavior arises here beyond what is already known for SubPcs, SubPors, or related contracted systems.

A key advance that distinguishes the present work from previous studies is the identification of a distinct protonation mode in globally non-aromatic tripyrrolic macrocycles, which leads to the formation of a tetracoordinated boronium cation rather than a borenium species. Although the boron–tripyrrolic framework itself may appear similar to previously reported systems, this work reveals that similarity at the structural level does not translate into equivalence in acid response, and that the newly identified protonation behavior is the origin of the superacid-resistance in the macrocyclic BODIPYs.

In particular, while the boron–calix[3]pyrrole complex was previously reported to retain the boron atom in trifluoroacetic acid, this earlier work described only the phenomenological acid tolerance, and the underlying protonation mechanism remained unexplored. Since our initial report on the synthesis of boron–C3P complexes in 2021, we have systematically investigated this behavior and identified a reversible, dearomative protonation pathway as the key stabilizing mechanism under acidic conditions. Elucidation of this mechanism directly enabled the rational design of the present superacid-resistant BODIPYs.

Accordingly, the current study does not represent a simple extension of prior boron–tripyrrole chemistry, but rather reveals a previously unrecognized synergy between boron coordination and tripyrrolic macrocycles, and demonstrates how this synergy can be exploited to achieve exceptional acid resistance in BODIPY chromophores.

To make this conceptual distinction explicit, we have revised the Discussion section to clearly articulate the mechanistic difference in protonation behavior and to emphasize how this insight underpins the conceptual advance of the present study.

4. The proposed “acid-induced fluorescence switching” mechanism—arising from suppression of intramolecular quenching—is a recurring concept in chromophore design and has been widely documented. The authors should better contextualize their findings within this literature and specify whether any mechanistic feature here is truly unprecedented.

The Rehm–Weller analysis of $\Delta G(\text{PET})$ indicates that photoinduced intramolecular electron transfer is thermodynamically favorable under neutral conditions (please also see our response to Reviewer 1 and Supplementary Fig. 26), supporting a PET-based fluorescence quenching mechanism that is conceptually similar to those widely employed in acid-responsive fluorescent sensors. We therefore agree that the basic “acid-induced fluorescence turn-on” mechanism itself is not unprecedented. However, in previously reported systems, such switching is typically limited to moderately acidic conditions, as strong acids often induce irreversible degradation of BODIPY chromophores.

In the present system, the key mechanistic distinction lies in the unique acid response of the macrocycle-embedded pyrrole unit, which proceeds via formation of a tetracoordinated boronium cation. This stabilization mode preserves the BODIPY framework even under

superacidic conditions, thereby enabling PET-based fluorescence switching in an acidity regime that has been inaccessible to conventional BODIPYs.

5. Limited optical tunability and modest spectral range: the emission remains confined to the 500–600 nm region. Extension into the near-infrared, which is the current frontier in bioimaging, would represent genuine novelty and impact. Similarly, the observed red-shifts upon peripheral aryl substitution are small (compared with subporphyrinoids or subporphyrazines (e.g., J. Org. Chem. 2020, 85, 1152)), and the substantial decrease in quantum yield upon functionalization raises questions about the practical value of these derivatives.

To demonstrate the optical tunability of the superacid-resistant BODIPYs, we added a new synthetic route to *meso*-aryl-substituted analogues using acyl chlorides, and summarized the optical data for peripherally aryl substituted analogues **15–19** in Table 2. As also discussed in our response to Reviewer 1, although the red-shift observed for compound **15** is modest, careful re-examination revealed that it retains a moderate fluorescence quantum yield (47%) in sulfuric acid. While rotation of the *meso*-aryl ring in the excited state accelerates non-radiative decay pathways, as commonly observed for conventional BODIPYs, intramolecular charge-transfer interactions between the *meso*-thienyl moiety and the BODIPY core resulted in a pronounced red-shift of the fluorescence spectrum, approaching the near-infrared region (ca. 750–800 nm) in highly acidic media. In addition, the fluorescence quantum yields of these analogues were enhanced upon restriction of intramolecular rotation of the aryl groups in viscous solvents or frozen solutions. Notably, *meso*-phenylated analogues **16–18** in frozen MSA solutions exhibited fluorescence quantum yields of 79–85%, which are comparable to that of compound **11**, despite their low quantum yields (<5%) in non-viscous acids.

The revised data demonstrate that macrocyclic BODIPYs retain optical tunability through peripheral substitution in a manner analogous to conventional BODIPYs. Importantly, the aryl-substituted analogues **15–19** did not undergo deboronation even in superacidic media, thereby enabling fluorescence tuning under conditions that are inaccessible to conventional BODIPY systems.

Additional Remarks

1. Literature coverage

Several seminal works are missing. For instance, the comprehensive Chem. Rev. 2007, 107, 4891–4932 on BODIPYs should be cited, as well as the most recent reviews on subporphyrinoids (Chem. Soc. Rev. 2022, 51, 9482; Trends Chem. 2023, 5, 279). The current bibliography underrepresents key prior art, weakening the contextual framing.

We thank the reviewer for this suggestion. We have added the recommended review articles as refs. 9, 20, and 21.

2. Line 227: “solubile” → “soluble”. Please also perform a careful language and consistency revision

We thank the reviewer for pointing this out. We have corrected “solubile” to “soluble” and have carefully revised the manuscript for language accuracy and consistency. In particular, the substrate for the synthesis of BODIPY **11** is not “triethyl orthoformate” but “trimethyl orthoformate”, and we have corrected this accordingly.

In summary, the manuscript presents technically competent work and interesting synthetic chemistry, but it falls short of the conceptual advance and broad significance expected for Nature Communications. The acid stability and post-functionalization are incremental improvements within an already mature class of compounds. A more comprehensive comparison with known acid-resistant fluorophores, clarification of the mechanistic novelty, and demonstration of distinctive optical performance (e.g., NIR emission, tunability) would be required for reconsideration.

We again appreciate the reviewer’s careful evaluation and the opportunity to address the conceptual advance of our work.

In the revised manuscript, (1) we explicitly articulate that dearomative protonation at a pyrrole unit leading to the formation of a boronium cation represents a previously unrecognized protonation mode in tripyrrole-based macrocyclic chemistry and constitutes a key mechanism underlying the observed superacid resistance, (2) the comparison of acid tolerance with existing BODIPYs clearly demonstrated the substantial expansion of the accessible acidity range for BODIPY dyes, and (3) implementation of an acyl chloride-based synthetic route to *meso*-aryl-substituted macrocyclic BODIPYs expanded the scope of derivatization and enabled further tunability of emission wavelengths through intramolecular charge-transfer interactions. We believe that these revisions clarify that the present work goes beyond an incremental extension of boron–tripyrrole chemistry and identifies a new mechanistic principle for designing superacid-resistant BODIPY chromophores.

Reviewer 4

This is a very interesting paper that describes the synthesis, characterization and application of superacid resistant cyclic BODIPYs. I believe the work presented is significant to the field of BODIPYs and their practical applications; a clear comparison between the new reported acid-stable cyclic BODIPYs and non-cyclic BODIPYs is investigated by the authors. All compounds are adequately characterized by NMR and HRMS. The methodologies used for the characterization of the neutral and positively charged species are all adequate and the conclusions are supported by the results obtained. The Figures are adequate and informative.

We thank the reviewer for the positive and encouraging evaluation of our work.

I suggest that the authors correct a few typos in the manuscript, such as in lines 227 and 248. In addition, the Discussion section is more of a Conclusions section.

We also thank for pointing this out. We have corrected the typographical errors:

For line 227: “solubile” → “soluble”

For line 248: “these” → “there”

We have also revised the Discussion section to clarify its purpose and structure.

Response to Editorial Requirements

- In accordance with the editorial policy, we have prepared a Source Data file containing the raw data underlying the figures and tables, and this has been uploaded with the revised manuscript.
- During the revision process, the author list was updated to reflect substantial contributions made to the additional experiments. The completed author list change approval form, signed by all authors, has been submitted.

Point-by-Point Response to the Reviewers' comments

(*blue*: original comments by reviewers, *black*: our response)

Reviewer 1

The authors have appropriately addressed the major concerns. However, to ensure reproducibility, the methodological details in the Supporting Information needs to be expanded in detail prior to acceptance. The newly conducted experimental procedures, such as phosphorescence and singlet oxygen measurements, were wrapped in the captions of Figures. They should have a more detailed experimental section. Other than that, this excellent manuscript is recommended for acceptance.

We thank the reviewer for re-evaluating our revised manuscript and for the positive assessment and recommendation for acceptance. To ensure reproducibility, detailed experimental procedures for the phosphorescence and singlet oxygen measurements have been added to the supporting information (page S52).

Reviewer 2

We also thank this reviewer.

Reviewer 3

The authors have clarified all the concerns raised by this referee. With these changes, this referee is now convinced that the work is suitable for Nat. Comm, and therefore recommends its publication.

We thank the reviewer for the careful re-evaluation of our manuscript and for the positive recommendation.

Response to Editorial Requirements

In accordance with the editorial checklist, we have revised the main manuscript and Supporting Information to comply with all formatting and presentation requirements. For ease of review, we have also provided versions of the revised manuscript and Supporting Information in which all changes are highlighted in yellow. No scientific content or data have been altered during this editorial revision.